# Prolonging Campaign Life of Blast Furnace Trough by Water Cooling

**DOI:** 10.3390/ma16030891

**Published:** 2023-01-17

**Authors:** Zhiyuan Li, Haifei Wang, Fengshou Ding, Huiqing Tang

**Affiliations:** 1State Key Laboratory of Advanced Metallurgy, University of Science and Technology, Beijing 100083, China; 2REWELL Refractoy Zhengzhou Company, Zhengzhou 450041, China

**Keywords:** blast furnace trough, refractory materials, water cooling, campaign life

## Abstract

A long campaign life of the trough of the blast furnace (BF) is important for improving the BF operation efficiency, reducing the cost of blast furnace ironmaking, and bettering the working conditions of the casting yard. In this paper, a water cooling device was designed for the trough of a BF with a volume of 630 m^3^. The effect of water cooling on lengthening the unit campaign life of the trough was investigated using industrial tests, numerical simulations, and theoretical analysis. Results showed that by using water cooling, the throughput of the trough in a unit campaign was increased by 40,000 tons, and its unit campaign life was increased by 34 days. During a unit campaign cycle, the influence of the water cooling device on the temperature distribution in refractory materials gradually developed from the low-temperature zone to the high-temperature zone, and the expansion of the high-temperature zone was suppressed. Therefore, the water cooling device inhibited the dissolution of refractory materials and retarded the chemical erosion from the molten slag and the atmosphere.

## 1. Introduction

In blast furnace (BF) ironmaking, the iron runner system is a device consisting of the main trough, an iron runner, a slag runner, and a skimmer. During BF tapping, the main trough plays the function of transporting and separating iron from slag after its extraction from the taphole. A long service of the trough is important to BF ironmaking as it can improve BF operation efficiency, reduce hot metal cost, and better the working environment of the tapping yard.

A typical campaign cycle of the trough could be described as follows: complete installation, operation, and demolition. During the campaign, frequent repairs are needed to maintain a satisfying performance of the trough. The repair of the trough owes to the degradation and wear of the lining refractory. The life of the trough depends on several factors including the BF operation conditions [1,2], the geometry of the trough [3,4,5,6], and the physicochemical properties of refractory materials [7,8]. Sustained efforts have been made in academia and the industry, and the trough life has been considerably lengthened [9]. Nowadays, a unit campaign life of the trough (trough life between two major repairs) reaches approximately two months (or in other words, its life capped at the throughput reaches 100,000 tons of hot metal (tHM) to 150,000 tHM, corresponding to a specific refractory consumption of approximately 0.6 kg/tHM [10]). The main method to lengthen the trough life is to improve the properties of refractory materials and, in this way, the development of Al_2_O_3_-iC-C castables can now fulfill many requirements of modern-day BF ironmaking practices [11]. However, further improving the quality of refractories faces many difficulties and high costs. Theories and experiments have disclosed that the performance of refractory materials depends closely on their working temperatures [12]. By decreasing the working temperature, the kinetics of various temperature-dependent processes such as chemical reactions, dissolution, etc. are retarded, thus reducing the wear rate of the trough lining. Presently, natural cooling or forced air cooling methods are adopted in lowering the temperature of the trough refractory materials [13,14]. In the iron and steel industry, heat dissipation of refractory materials is usually enhanced by water cooling. This has been demonstrated by several metallurgical reactors. In BFs, the water cooling hearth [15,16] and the water cooling tuyere [17] are employed. In submerged-arc furnaces, the “freeze lining” refractory system is adopted [18,19], and, in electric arc furnaces, the water cooling panel and the water cooling roof are used [20,21]. As a holding vessel of hot metal and slag, it is also expected that water cooling can enhance the heat dissipation of the trough refractory materials, and thus the campaign life of the trough can be further lengthened. However, no water cooling device has been applied in the trough up to now.

In this research, a water cooling device for the BF trough was designed. Industrial tests were then carried out to examine its influence on the trough life. Thereafter, the influence of water cooling on the temperature field in the trough was numerically investigated and the mechanism of water cooling for protecting the trough was analyzed.

## 2. Design of Water Cooling Device and Industrial Tests

The object for the present study was the No. 1 BF of the Xinchuang iron and steel company in Yunnan Province (China). As the BF operating parameters exert significant influences on the trough life, the employed BF was introduced briefly first. The BF had an effective volume of 630 m^3^, producing 2400 tHM/day. The sinter used in the BF was of low grade, resulting in a large slag rate of approximately 400 kg/tHM. The temperature of HM and slag fluctuated around 1753 K. The composition of the slag is shown in Table 1. The BF cast iron and slag had two troughs. Each trough had a daily HM throughput of approximately 1200 tHM. Only one trough was selected for the present research. The trough was 14.4 m long and 2.3 m wide. The trough was formulated by three distinct layers and they were working layer, permanent layer, and steel frame. The working layer was in direct contact with the iron and slag. The permanent layer was in the lower part of the trough. The refractories for the working layer and the permanent layer were made of Al_2_O_3_-SiC-C castables. Chemical compositions of castables in the working layer and the permanent layer are listed in Table 2.

A water cooling device was designed for the trough. The water cooling device was composed of several water cooling plates. The structure of a water cooling plate is shown in Figure 1. The cooling plate was made of steel. The plate had a length of 2.0 m and a width of 0.6 m. Each plate had a water inlet and a water outlet. The water source for the water cooling plates was supplied by the BF cooling system. The installation pattern of the water cooling device on the trough is shown in Figure 2. Six water cooling plates were employed and they were intimately attached to the steel frame. On each side, there were three plates.

The industrial tests were conducted on the trough with no water cooling first. In a unit campaign cycle, the top thicknesses of both embankments (north embankment and south embankment in Figure 2) of the trough were measured every 7 days. The measured position was 3.5 m from the taphole (plane D in Figure 2), which was near the entry region of the iron–slag jet. After a unit campaign cycle, the water cooling device was installed on the outside of the steel frame and the same test procedures were repeated.

## 3. Numerical Simulation

The temperature field in the refractory materials of the trough was studied using numerical simulations. Nowadays, the trough is constructed in a pooling type to minimize the thermal shock on its refractory materials. This means that, in BF casting, the trough fills with molten iron and slag, and after the taphole is plugged with clay materials some iron and slag are kept in the trough. Therefore, the trough temperature can quickly reach a steady state after a certain number of castings. Considering these facts, a two-dimensional heat transfer model on a cross-section of the trough was developed. The simulated cross-section is Section E in Figure 2, which is located in the middle of the water cooling plate B. Since the trough is almost symmetrical in respect to its longitudinal axis, the computational domain is half of Section E. Because the trough geometry (such as the floor slope, width, and wall angle) is gradually altered during a unit campaign cycle, in Section E, four cases, namely Cases I, II, III, and IV, were investigated. Cases I and II were for the trough in the initial stage of a unit campaign cycle, and Cases III and IV were for the trough at the end of a unit campaign cycle.

Case I was for the trough without water cooling, and Case II was for the trough with water cooling. The modeling geometry and its subdomains and boundaries are represented in Figure 3. The computational domain represented by Ω is subdivided into three subdomains, and the boundary of Ω, represented by Γ, is subdivided into eight sub-boundaries.

Ignoring the internal heat sources, the heat conduction equation for heterogeneous isotropic materials in Ω is Equation (1).
(1)$$\frac{\partial }{\partial x}(k\frac{\partial T}{\partial x})+\frac{\partial }{\partial y}(k\frac{\partial T}{\partial y})=0$$
where *k* is the thermal conductivity of the refractory material, W/(m·k); *T* is the temperature, K; *x*, *y* are the two-dimensional coordinate directions, m.

The thermal conductivity of the material in each subdomain is determined according to the following method. Due to the oxidation of carbon particles in the working layer of the trough, porosity and pore size in the castable become large, resulting in a sharp decrease in its thermal conductivity. Therefore, the thermal conductivity of the working layer (Ω_1_) is assumed to be 1.0 W/(m·k). In the permanent layer (Ω_2_), since the castable in the permanent layer is insignificantly eroded and is continuously pressed by molten iron and slag, it becomes dense. Therefore, its thermal conductivity is increased and is assumed to be 3.0 W/(m·k). The thickness of the steel frame (Ω_3_) is 15 mm, and its thermal conductivity is taken as 54 W/(m·k).

The boundary conditions of Case I are the following. On the symmetry ($\Gamma_{1}$ and $\Gamma_{2}$), no heat flux exists. The temperature of the hot side ($\Gamma_{3}$) is considered to be constant and is equal to the temperature of molten iron and slag (1753 K) because the hot side is exposed to them. Regarding the top surface ($\Gamma_{4}$), no heat exchange is considered as the cooling effect of natural air convection is compensated by the exothermic reaction of carbon oxidation. On the lateral sides ($\Gamma_{5}$, $\Gamma_{6}$, $\Gamma_{7}$) and the bottom side ($\Gamma_{8}$), heat transfer by natural air convection is assumed and is described using Equation (2).
(2)$$k\frac{\partial T}{\partial n}=h(T_{\mathrm{air}}-T)$$
where $\vec{n}$ is the outer unit normal vector, *h* is the heat transfer coefficient and is 5.0 W/(m^2^·K) [13], and *T*_air_ is the temperature of the atmosphere and is 313 K.

The boundary conditions of Case II are similar to those of Case I except for that on $\Gamma_{6}$. In Case II, the heat flux on $\Gamma_{6}$ is calculated using Equation (3), which is based on the cooling water flow rate in the water cooling device, and the temperature difference in water between the inlet and the outlet.
(3)$$F_{\mathrm{heat}}=(T_{in}-T_{out})*Cp*Q/A$$
where *F*_heat_ is the heat flux, J/(m^2^·s); *T_in_* and *T_out_* are the temperatures of the inlet and outlet water of the water cooling plate, K, respectively; *Cp* is the specific heat capacity of water, J/(kg·k); *Q* is the water flow rate, kg/s; and *A* is the contacting area between the cooling plate and the steel frame, m^2^.

In the computation domain, the above model was numerically solved using an in-house code developed using the C programming language. In the numerical simulations, the finite volume method (FVM) and structure grid method [22] were used. The area of each grid in the computational domain is less than 1.0 cm^2^, and the boundary conditions are considered as source items in the discrete equation of Equation (1). The criterion for convergence is Equation (4).
(4)$$\sum \left|T_{new}-T_{old}\right|/\sum T_{old}<10^{-6}$$
where *T_new_* is the new iteration temperature of the grid, and *T_old_* is the old iteration temperature of the grid.

Regarding Cases III and IV, the geometry of the computational domain is shown in Figure 4. To ensure the safe operation of the trough, the top thicknesses of both embankments should be larger than 300 mm. The computational domain in Figure 4 includes three subdomains (Ω_1_, Ω_2_, and Ω_3_), and the boundary includes eight sub-boundaries ($\Gamma_{1}$,$\Gamma_{2}$…$\Gamma_{8}$). Case III is without water cooling and Case IV is with water cooling. The solution methods of Cases III and IV are similar to those of Cases I and II, respectively.

## 4. Results and Discussion

### 4.1. Results of Industrial Tests

Figure 5 shows the results of industrial tests of the north embankment. It could be observed that, in Figure 5, after installing the water cooling device, the top thickness decrease became notably slow. The results of industrial tests of the south embankment are shown in Figure 6 and a similar trend could be observed. Therefore, the water cooling device can weaken the degradation of the refractory lining of the trough. From Figure 5 and Figure 6, it can be seen that the accumulated HM throughput in a unit campaign cycle of the trough increased from 80,000 tHM to 120,000 tHM after installing the water cooling device. The daily throughput is about 1200 tHM. Therefore, the trough life increased from 66 days to 100 days by installing the water cooling device.

The above findings show that using water cooling has a distinguished effect on lengthening the trough life.

### 4.2. Effect of Water Cooling on Temperature Field in Trough Refractories

In the industrial tests with the water cooling device, for all watering cooling plates, the water flowing rate was 2.5 t/h, and it was observed that the water temperature difference between the inlet and the outlet was approximately 10 K. Therefore, according to Equation (3), the cooling intensity of the water cooling device on the trough was 24,000 W/m^2^. Simulation results of Cases I and II are shown in Figure 7a,b. Compared to Figure 7a, the isothermal line of 773 K in Figure 7b moves towards the symmetry, indicating that in the initial stage of a unit campaign cycle, the water cooling device can enlarge the area of the low-temperature zone. However, compared to Figure 7a, the positions of the isotherm 1373 K in Figure 7a,b do not show a considerable difference, reflecting that their effect on the high-temperature zone is insignificant.

Simulation results of Case III and Case IV are shown in Figure 8a,b. Compared to Figure 8a, the area between the isotherm line 1373 K and 1673 K in Figure 8b becomes smaller, reflecting that the high-temperature zone was narrowed by the water cooling near the end of a unit campaign cycle. Therefore, during the service of the trough, the water cooling device inhibits the development of the high-temperature zone (*T* > 1373 K) to the steel frame.

The simulation results disclosed that the water cooling device had an insignificant influence on the area of the high-temperature zone of the trough in the initial stage of a unit campaign cycle. However, it suppressed the expansion of the high-temperature zone during the service of the trough.

### 4.3. Mechanism

As can be seen from Table 2, the main components in the castable of the working layer are Al_2_O_3_, SiC, C, and metallic silicon, and the corundum aggregate occupies a large portion. In addition, the pitch is mixed in as an additive. In the baking stage of the trough, the pitch is carbonized and then the graphite carbon is precipitated, providing a binding force for the particles in the castable. During the BF casting, the periodic damages to the lining are dissolution into the slag and chemical attack from atmosphere, molten slag, and molten iron.

The iron–slag jet from the taphole enters in the front of the trough, scouring both the hot side and the bottom of the trough. In the region between the slag line and the metal line, dissolution of the castable occurs after it contacts the slag. However, this erosion is negligible below the metal line. The mechanism of this erosion is that the molten slag penetrates the hot side through pores. SiO_2_ and CaO in the slag react with Al_2_O_3_ in the lining, forming some low-melting-point compounds including anorthite (CAS2) and gehlenite (C2AS) under the reactions given by Equations (5) and (6). The temperature range of slag solidification is from 1373 K to 1673 K [23]. The calculation of Gibbs free energies of the reactions given by Equations (5) and (6) under standard state [24] showed that C2AS and CAS2 could be easily formed under temperatures above 1373 K. These compounds were then gradually dissolved into the slag. Hence, the small Al_2_O_3_ particles leave the lining with the flowing slag. After the small Al_2_O_3_ particles in the lining are depleted, the dissolution of slag shifts to corundum aggregate. Once the corundum-aggregate-based structure is destroyed, the SiC particles begin to loosen and are flushed into the slag.
$${\mathrm{CaO}+\mathrm{Al}}_{2}O_{3}{+2\mathrm{SiO}}_{2}=\mathrm{CaO}\cdot {\mathrm{Al}}_{2}O_{3}\cdot 2{\mathrm{SiO}}_{2}$$
(5)$$\Delta G^{0}=-97.99-0.025T\;(\mathrm{kJ})$$
$${2\mathrm{CaO}+\mathrm{Al}}_{2}O_{3}{+\mathrm{SiO}}_{2}=2\mathrm{CaO}\cdot {\mathrm{Al}}_{2}O_{3}\cdot {\mathrm{SiO}}_{2}$$
(6)$$\Delta G^{0}=-139.32-0.020T\;(\mathrm{kJ})$$

From Figure 7 and Figure 8, it is known that after the water cooling device is installed, with the expansion of the hot side of the trough, the high-temperature zone (*T* > 1373 K) in the refractory is continuously decreased. The viscosity of the slag increases with the decrease in temperature [25]. Therefore, by installing the water cooling device, the penetrating area of the slag in the lining is narrowed. This means that the water cooling device reduces the zone where the slag and the castable can make contact. Furthermore, even though the slag can reach Al_2_O_3_ particles through pores to generate the low-melting compounds, the temperature in the refractory material is lower than their melting points. Thus, these low-melting compounds cannot be taken away by the slag in a molten state. The above analysis shows that the water cooling device can weaken the flushing effect of high-temperature slag on the Al_2_O_3_-based matrix of the trough.

The chemical erosions of the main trough are solely from the atmosphere and the molten slag. These erosions occur above the slag line of the trough. During the cessation of cast operation, part of the slag/lining interface is exposed to the atmosphere, and during cast operation, some air is entrained in the iron–slag jet. The molten iron is oxidized by O_2_ in the atmosphere following the reaction given by Equation (7). SiC in the lining then reacts with FeO following the reactions given by Equation (8). O_2_ in the atmosphere reacts with the carbon particles according to the reaction given by Equation (9), and SiO_2_ and FeO in the slag react with carbon particles following the reactions given by Equations (10) and (11), respectively. From the Gibbs free energies of the reactions given by Equations (7)–(10) in the standard state [24], it is known that these reactions can proceed on the slag/lining interface as the interface has a high temperature above 1373 K. Regarding the reaction given by Equation (11), its Gibbs free energy in the standard state is positive under a temperature above 1373 K. However, as the partial pressure of gaseous SiO is very low, it can also be carried out on the slag/lining interface. O_2_ in the atmosphere diffuses through the pores of the lining to react with carbon forming many channels in the lining. These channels facilitate the permeation of slag into the lining leading to a decarburized layer in the vicinity of the slag/lining interface. As the binging force in the decarburized layer is weak, the lining is gradually spalled and damaged. A small ratio of some antioxidants (e.g., aluminum powder and metallic silicon powder) is added to the castable in the working layer to prevent carbon oxidation. Under high temperatures, they can preferentially react with oxygen forming high-temperature ceramic bonds to inhibit carbon oxidation to some extent. However, due to the long service time of the trough, the occurrence of the reaction (7)–(11) is still inevitable after the main trough is used for a certain period.
$${2\mathrm{Fe}(l)+O}_{2}\left(g)=2\mathrm{FeO}(l\right)$$
(7)$$\Delta G^{0}=-504.6+0.114T\;(\mathrm{kJ})$$
$${\mathrm{SiC}(s)+2\mathrm{FeO}(l)=2\mathrm{Fe}(l)+C(s)+\mathrm{SiO}}_{2}\left(l\right)$$
(8)$$\Delta G^{0}=-314.4+0.042T\;(\mathrm{kJ})$$
$${2C(s)+O}_{2}\left(g)=2\mathrm{CO}(g\right)$$
(9)$$\Delta G^{0}=-224.6-0.176T\;(\mathrm{kJ})$$
FeO(l)+C(s)=Fe(l)+CO(g)
(10)$$\Delta G^{0}=138.3-0.144T\;(\mathrm{kJ})$$
$${\mathrm{SiO}}_{2}\left(l)+C(s)=\mathrm{SiO}(g)+\mathrm{CO}(g\right)$$
(11)$$\Delta G^{0}=672.04-0.332T\;(\mathrm{kJ})$$

The oxidation of the carbon particles (reaction given by Equation (9)) is active above 1073 K but is negligible below 1073 K [26]. From Figure 7 and Figure 8, it is known that after the water cooling device is installed in the trough, the expansion of the zone above 1073 K in the lining is suppressed. Therefore, carbon oxidation can be considerably retarded. In addition, due to the inhibition of carbon oxidation, the number of channels in the castable is greatly reduced, inhibiting the penetration of molten slag into the working layer. Therefore, reactions given by Equations (8), (10), and (11) can only occur on the lining/slag interface, but cannot develop into the interior of the lining. The resistance of the working layer to the erosion from FeO and SiO_2_ in the slag is thus enhanced.

## 5. Conclusions

A water cooling device has been designed for a BF with a volume of 630 m^3^. The effect of water cooling on lengthening the campaign life of the trough was investigated using industrial tests, numerical simulations, and theoretical analysis. The following conclusions were drawn.

(1)The industrial tests showed that, after installing the water cooling device, the throughput of the main trough increased from 80,000 tons to 120,000 tons, and its unit campaign life increased from 66 days to 100 days.(2)The simulation results disclosed that the influence of the water cooling device on the temperature distribution in the refractory material of the trough gradually develops from the low-temperature zone to the high-temperature zone in a unit campaign cycle, inhibiting the expansion of the high-temperature zone in the working layer.(3)Theoretical analysis indicated that the water cooling device inhibited the dissolution of the castable and retarded the chemical erosion from molten slag and the atmosphere.

## Figures and Tables

**Figure 1 materials-16-00891-f001:**
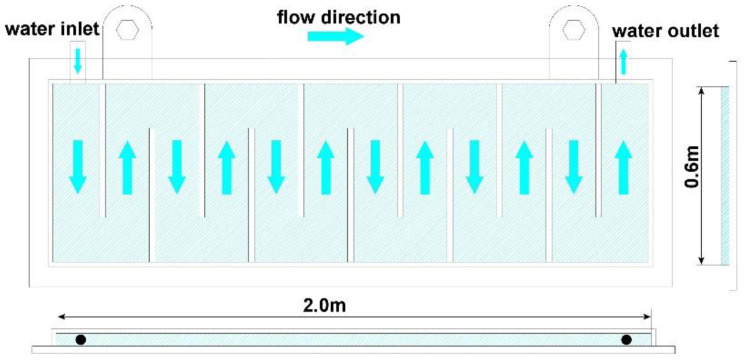
Structure of the water cooling plate.

**Figure 2 materials-16-00891-f002:**
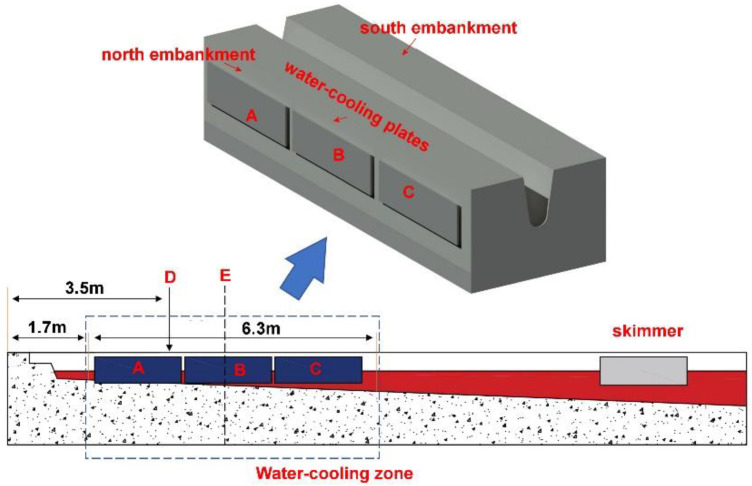
Illustration of the installation pattern of the water cooling device on the trough (A, B, and C: the three water-cooling plates on the north embankment of the trough; D: the measured plane on the trough; and E: the selected plane for modeling).

**Figure 3 materials-16-00891-f003:**
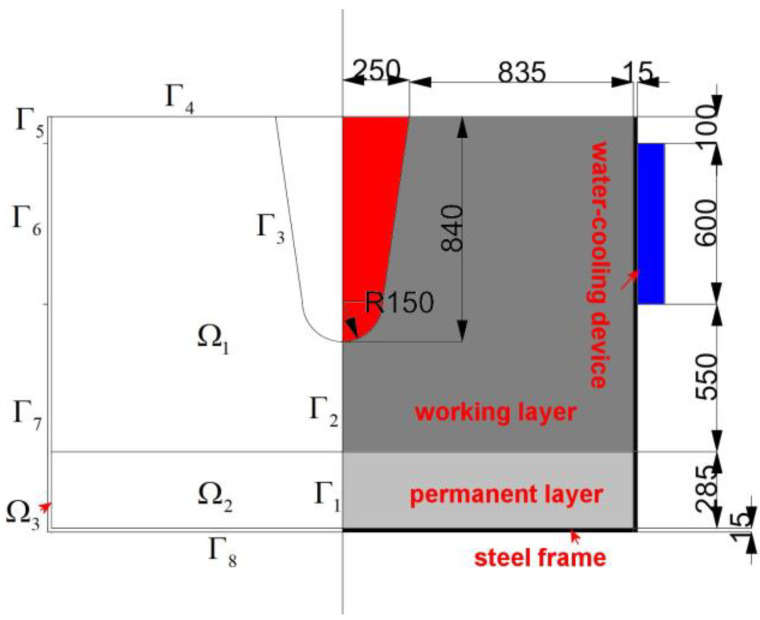
Computational domain and schematic diagram of the cross-section in the trough in the initial stage of a unit campaign cycle (unit: mm).

**Figure 4 materials-16-00891-f004:**
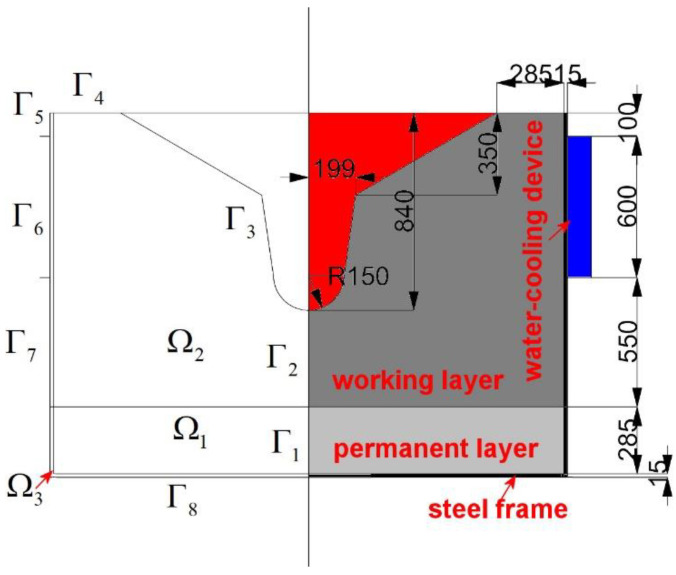
Computation domain and schematic diagram of the cross-section in the trough near the end of a unit campaign cycle (unit: mm).

**Figure 5 materials-16-00891-f005:**
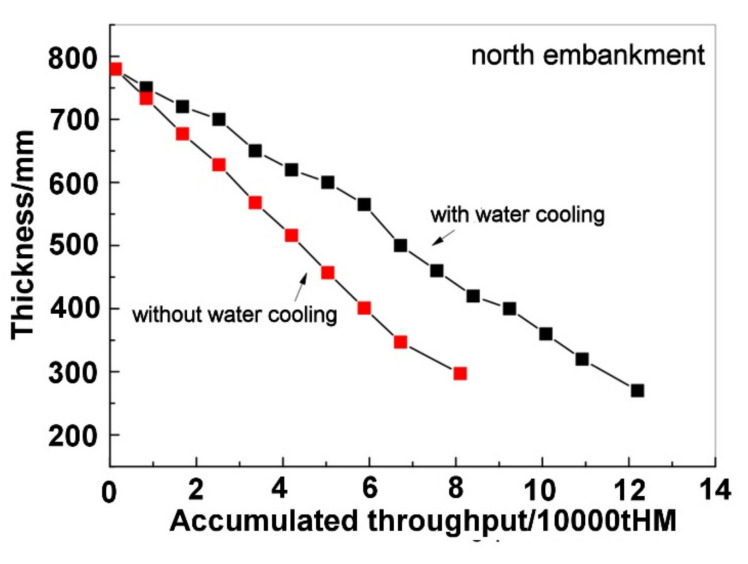
Relation between the top thickness of the north embankment and the accumulated throughput.

**Figure 6 materials-16-00891-f006:**
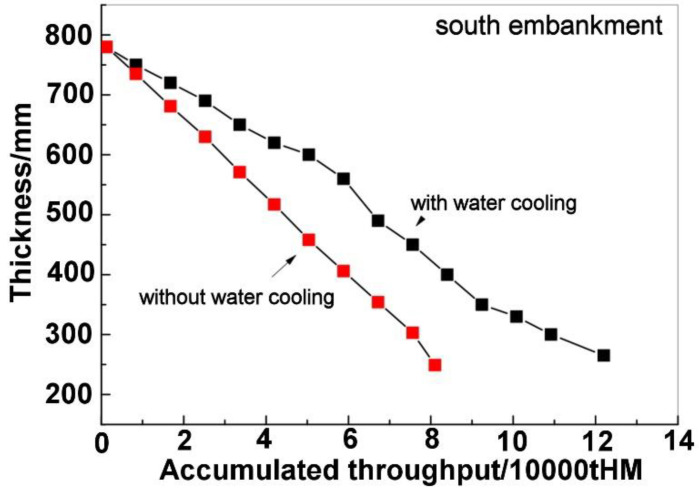
Relation between the top thickness of the south embankment and the accumulated throughput.

**Figure 7 materials-16-00891-f007:**
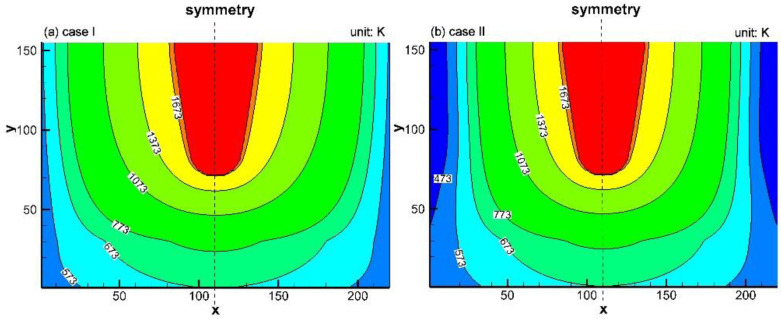
Temperature distribution in the trough materials in the initial stage of a unit campaign cycle: (**a**) without the water cooling device and (**b**) with the water cooling device.

**Figure 8 materials-16-00891-f008:**
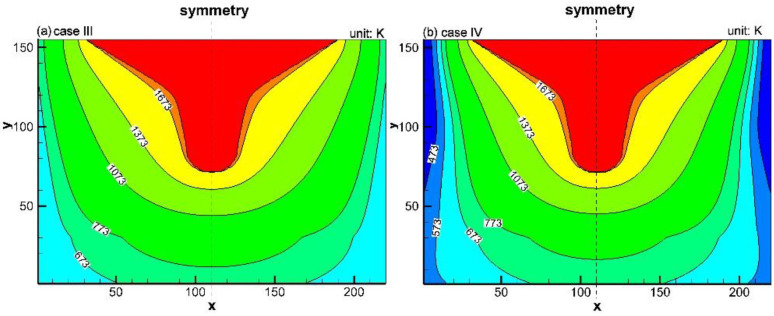
Temperature distribution in the trough materials near the end of a unit campaign cycle: (**a**) without the water cooling device and (**b**) with the water cooling device.

**Table 1 materials-16-00891-t001:** Composition of the blast furnace slag (wt*%*).

Component	SiO_2_	CaO	Al_2_O_3_	MgO	MnO	S	FeO	TiO_2_
Content	32	37	13	8	0.62	0.73	0.62	1.86

**Table 2 materials-16-00891-t002:** Composition of the castable in the working layer of the trough (wt%).

Component		SiC (98)	Corundum	Alumina Powder	Silica Fume	Cement (70)	Additive
Content	Working layer	35	47.5	6	4	5	2.5
	Permanent layer	7	75.5	6	4	5	2.5

## Data Availability

Not applicable.

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
