# Peer review of "Prolonging Campaign Life of Blast Furnace Trough by Water Cooling"

_materials, 2023, doi:10.3390/ma16030891_

Round 1

Reviewer 1 Report

Journal: Materials (ISSN 1996-1944)

Manuscript ID: materials-2017695

I have read the article "Prolonging campaign life of blast furnace trough by water cooling." The authors conducted industrial tests, numerical simulations, and theoretical analyses to increase blast furnaces' efficiency. When the literature is examined, I believe that the study is original. I think the work is suitable for the journal "Materials." However, the following major corrections are required for the article's publication.

1.      Although many studies on the cooling of blast furnaces exist in the literature, the authors did not mention this. However, in the introduction, the originality of the research and its difference from other studies should be clearly stated.

2.      According to which parameters was the selected cooling layer design prepared for industrial cooling. What is the difference between the cooler plates on the market?

3.      Results and discussion and conclusion parts are inadequate according to citation and analyze in detail. There should be the importance of the study in detail, comparison of results with other approaches in the literature, and the success of the experimental results.

4.      Improve the results and discussion and conclusion parts.

5.      The article should be rearranged by taking into account the journal writing rules and citation rules.

*** Authors must consider them properly before submitting the revised manuscript. A point-by-point reply is required when the revised files are submitted.

Reviewer 2 Report

This manuscript focus on the way to improve the campaign life of BF trough by using water cooling.

The English language of manuscript is a rather good. But there are some improvements needed:

1) The abstract and conclusions are almost similar. The authors should make these more concise and details.

2) The modelling/simulation methods and softwares used should be in details in 3 numeric simulation section.

3) In the 4.3. mechanism section, there are no cited references for the authors' statements: what is the conditions (composition range, temperature range, potential, or enthalpy of reaction...) for reactions (5)-(6) for slag formation; reaction (7)-(11) for both oxidation and reduction reaction.

Reviewer 3 Report

The document presented by the authors, in my opinion fits as Communication.

The theme is interesting, the problem is well structured and presented, however, we are in the presence of an industrial application to solve a specific problem. I have doubts about the scientific framework that justifies this publication in a scientific journal. The work presented here will be more suitable with a communication in a congress or something of that sort.

It should be noted that the framing of the theme carried out in the introduction chapter is very summary and, in the part, referring to the presentation of results, the component of critical analysis and mainly benchmarking is missing.

I emphasize again that in my opinion this work would be more suitable for a communication in a congress or something equivalent.

Round 2

Reviewer 1 Report

The authors made the desired corrections. In my opinion, this article can be accepted for publication in the "Materials" journal in its final form.

Reviewer 2 Report

The manuscript has been revised to make it more concrete and easy to follow.

The symbol for free energy should be corrected as ΔG°.

Reviewer 3 Report

 No comments